# Learning Quantifiable Visual Explanations Without Ground-Truth

## Abstract

Explainable AI (XAI) techniques are increasingly important for the validation and responsible use of modern deep learning models, but are difficult to evaluate due to the lack of good ground-truth to compare against. We propose a framework that serves as a quantifiable metric for the quality of XAI methods, based on continuous input perturbation. Our metric formally considers the sufficiency and necessity of the attributed information to the model's decision-making, and we illustrate a range of cases where it aligns better with human intuitions of explanation quality than do existing metrics.

To exploit the properties of this metric, we also propose a novel XAI method, considering the case where we fine-tune a model using a differentiable approximation of the metric as a supervision signal. The result is an adapter module that can be trained on top of any black-box model to output causal explanations of the model's decision process, without degrading model performance. We show that the explanations generated by this method outperform those of competing XAI techniques according to a number of quantifiable metrics.

## 1 Introduction

Recent deep learning (DL) models have achieved outstanding success in a range of predictive tasks, especially those that have historically been out of reach of conventional statistical models and pattern recognition systems with hand-engineered features. However, this success has come at the cost of model interpretability. DL models are often called "black boxes" due to their complex and multivariate decision processes, which cannot be directly explained on a human level. In many applications this is not a critical weakness, but it poses an obstacle to the deployment of such systems where accountability and transparency are key considerations. This challenge only grows as AI systems scale up, both in terms of model architecture and widespread adoption.

Historically, a range of **explainable AI (XAI)** techniques have been used to provide human-interpretable explanations for the output of DL models (Ras et al., 2022). In the visual domain, these often take the form of pixel-level saliency maps that estimate the contribution of each input pixel to the model's decision process. Such methods are widely used in debugging visual models during training, to ensure that the patterns they are learning represent real semantic features and are not overfitting to spurious data correlations.

Yet despite the availability of sophisticated techniques for explaining model outputs, it remains difficult to adequately *quantify* the suitability of any given explanation (Tomsett et al., 2020). An open challenge in the XAI research field is how to appropriately rank explanations against each other, given a model and its output on some example of interest. The field lacks consensus on an appropriate metric for explanation quality. This challenge is partly due to the lack of meaningful ground-truth, and partly to how explanations serve diverse, often conflicting purposes in the ML pipeline. Unlike predictions at inference time, which can be objectively validated, explanations exist primarily for human use. Their usefulness depends on context and specific use case - whether debugging, auditing or supporting human-led decisions. As such, what makes a "good" explanation varies across applications, which has contributed to the lack of a standardized evaluation metric.

Of the methods that exist to quantify the quality of model explanations, the most prominent rely on perturbation of the input example according to local saliency scores (Alvarez Melis & Jaakkola,

2018). However, we find that the scores given by such metrics often do not align with intuitive notions of explanation quality - for example, by giving high scores to explanations that cover large, irrelevant regions, or that focus on specific, narrow features.

In this paper, we propose the definition of a new perturbation-based metric called **Minimality-Sufficiency Integration (MSI)**, grounded in the information bottleneck framework (Tishby & Zaslavsky, 2015), designed to favour explanations that are simultaneously specific and parsimonious.

We also introduce a novel XAI technique that generates explanations according to the requirements specified by the above metric. This method, called **Learnable Adapter eXplanation (LAX)**, consists of training a self-supervised explanation module over an existing pre-trained model, without ground-truth annotations. We discuss the implementation of this approach in later sections.

## 2 RELATED WORK

### 2.1 EXPLAINABILITY IN VISION MODELS

XAI methods can be broadly divided into three categories (Ras et al., 2022): visualization-based approaches, distillation approaches, and intrinsic approaches. In the visual domain, visualization approaches are the most commonly used (Schulz et al., 2020; Selvaraju et al., 2017). The generated saliency map explanation represents a matrix of relevance scores associated to the input pixels, intended to give higher scores to relevant pixels and lower scores to irrelevant ones. The most common approaches to generate these saliency maps are gradient-based, for example, Grad-CAM (Selvaraju et al., 2017), LRP (Bach et al., 2015), and DeepLIFT (Shrikumar et al., 2017). The shared idea of these approaches is to calculate saliency maps by propagating gradients or relevance scores from the output layer back through the network to estimate the contribution of each input feature. However, they are susceptible to gradient shattering (Balduzzi et al., 2017), which can make their attributions inconsistent or unstable.

Other approaches are perturbation-based, where the idea is to observe changes in the output by perturbing different regions of the input image, thus inferring the importance of each region based on the impact of its perturbation. Examples include occlusion sensitivity (Zeiler & Fergus, 2014), LIME (Ribeiro et al., 2016) and RISE (Petsiuk, 2018). While these approaches overcome the gradient shattering problem, they are computationally expensive, as they require evaluating a large number of perturbed input images for each explanation.

### 2.2 LEARNING TO EXPLAIN

An alternative to post-hoc methods is to learn explanations during the model training process (Ras et al., 2022). Thus, the vision model is explicitly trained to generate saliency map explanations alongside its predictions, allowing the process to be more efficient and faithful (Bang et al., 2021; Alvarez Melis & Jaakkola, 2018). These models generate the explanation in a single pass, avoiding the need for repeated perturbations at inference time. However, since most datasets lack ground-truth saliency maps, these models often rely on auxiliary supervision or priors to guide the explanation learning process (Choi et al., 2024; Souibgui et al., 2025).

### 2.3 QUANTIFYING VISUAL EXPLANATIONS

As discussed, evaluating explanation quality remains challenging due to the lack of ground-truth saliency maps in most datasets (Zhou et al., 2022). A common evaluation strategy is fidelity-based testing (Alvarez Melis & Jaakkola, 2018; Tomsett et al., 2020). Under this framework, removing important (high-saliency) pixels should degrade the model's performance, while removing unimportant (low-saliency) pixels should have minimal impact. This can be done in two directions: MoRF (Most Relevant First) and LeRF (Least Relevant First), which progressively perturb the most and least relevant regions, respectively; and by deletion (from full image) or insertion (from a blank image),

However, direct pixel perturbation can introduce a domain shift from the original training distribution, potentially confounding the results. To mitigate this, ROAR (Hooker et al., 2019) was proposed, which retrains the model from scratch on data where specific pixels have been removed.

While this reduces the impact of domain shift, it is computationally expensive, introduces new biases, and operates only on the dataset level without any ability to quantify individual explanations (Rong et al., 2023). To address these limitations, alternative metrics such as R-fidelity and F-fidelity (Zheng et al., 2024; 2025) have been proposed. These methods perturb only partial sets of relevant and irrelevant pixels and may include optional fine-tuning to preserve model stability, as does IDSDS (Hesse et al., 2024). As a result, they offer improved robustness to domain shift and reduced reliance on retraining. However, they nonetheless rely on altering the model that is to be explained, and face problems distinguishing between competing maps (see below).

## 3 METHODOLOGY

### 3.1 PROPOSED METRIC

We observe that existing perturbation-based fidelity metrics are prone to two key limitations:

**(i) Handling large masks.** As shown in Table 1, we consider two possible explainability heatmaps over the same input image. In both cases, feeding only the relevant pixels (in red) to the model produces a correct prediction. However, the second mask highlights more information than necessary, including features clearly not used for prediction. Both masks have near-identical scores according to insertion and deletion metrics, failing to reward the more compact and focused explanation.

Table 1: Comparison of two valid masks on the same image from the Synthetic-MNIST dataset according to several quantitative metrics, including our proposed MSI.

| Visual | Insertion (Pixel Value) (Morf) | Deletion (Pixel Value) (Morf) | Insertion (%) (Morf) | Deletion (%) (Morf) | Fid- (Insertion) (Pixel Value) | Fid+ (Deletion) (Pixel Value) | Fid- (Insertion) (%) | Fid+ (Deletion) (%) | Base Score ($\alpha_{\min} = 0.5$) | Mask Penalty | MSI Score |
|---|---|---|---|---|---|---|---|---|---|---|---|
| Small Mask | 0.981 | 0.118 | 0.889 | 0.164 | 0.018 | 0.876 | 0.109 | 0.83 | 0.864 | 0.247 | 0.617 |
| Big Mask | 0.986 | 0.118 | 0.889 | 0.164 | 0.011 | 0.876 | 0.109 | 0.83 | 0.868 | 0.887 | -0.019 |

**(ii) Handling multiple plausible explanations.** Figure 1 demonstrates a scenario where both the relevance mask and its complement produce correct predictions when applied to the same input, due to the presence of more than one valid explanation for the prediction. Metrics like MoRF-Deletion implicitly assume that there is a single correct explanation. Here, such metrics give a poor score to this mask, even though the mask correctly captures one of several valid attribution pathways.

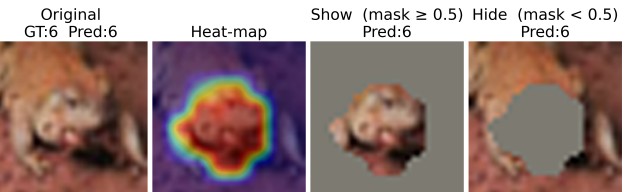

Figure 1: Example illustrating multiple valid solutions. From left to right: original image, full mask, thresholded mask region (values $\geq 0.5$), and its complement (values $< 0.5$).

We propose a single robust metric that addresses both limitations by balancing the minimality and sufficiency of proposed attributions to a model's output, incentivizing compact explanations and robustness to the presence of multiple informative regions. We name this metric **Minimality-Sufficiency Integration (MSI)**.

**Minimality-Sufficiency Integration.** MSI operates over an attribution heatmap normalized to the range $[0, 1]$, provided as an explanation for the output of a model on a given input example. It is computed as the sum of a base score and a mask size penalty.

The base score is defined as:

$$\text{BaseScore}_{\text{avg}} = \frac{1}{2} \Big[ \big( \text{Show}(> \alpha_{\min}) - \text{Show}(< \alpha_{\min}) \big)$$
$$+ \big( \text{AUC}_{\text{show}}^{\text{avg}} - \text{AUC}_{\text{hide}}^{\text{avg}} \big) \Big] \tag{1}$$

The threshold parameter $\alpha_{\min} \in [0, 1]$ binarizes the heatmap into "*important*" and "*less important*" regions. The first term compares predictive performance when masking these regions:

- $\text{Show}(> \alpha_{\min})$: Prediction accuracy when retaining only pixels with heatmap values greater than $\alpha_{\min}$.

- $\text{Show}(< \alpha_{\min})$: Prediction accuracy when retaining only pixels with values less than or equal to $\alpha_{\min}$ (i.e., the complement region).

A greater difference between these two scores indicates stronger separation of important and unimportant regions. This captures the case where multiple plausible explanations exist by pushing toward a correct separation between all the relevant regions and the non-relevant pixels.

The second term is based on area under the perturbation curve:

$$\text{AUC}_{\text{show}}^{\text{avg}} = \frac{1}{\Delta \alpha} \int_{\alpha_{\min}}^{1.0} \text{Show}(> \alpha) \, d\alpha \tag{2}$$

$$\text{AUC}_{\text{hide}}^{\text{avg}} = \frac{1}{\Delta \alpha} \int_{\alpha_{\min}}^{1.0} \text{Hide}(> \alpha) \, d\alpha \tag{3}$$

where

$$\Delta \alpha = 1.0 - \alpha_{\min} \tag{4}$$

- $\text{Show}(> \alpha)$ is the prediction accuracy when only the pixels with heatmap values greater than $\alpha$ are retained. This process is analogous to MoRF-Insertion, where pixels are progressively inserted in decreasing order of importance according to the heatmap values—for example, by revealing all pixels above 0.98, then 0.96, and so on. However, instead of continuing all the way to $\alpha = 0$, the integration is truncated at $\alpha_{\min}$.

- $\text{Hide}(> \alpha)$ is the prediction accuracy when pixels with heatmap values below $\alpha_{\min}$ are first removed, and then the remaining pixels are progressively deleted in order of decreasing importance, i.e., from values close to 1 down to $\alpha_{\min}$. This is analogous to (truncated) MoRF-Deletion.

To ensure fairness across different threshold settings, both AUC terms are normalized by the integration range $\Delta \alpha$. Without this normalization, lower thresholds would cover a larger interval, potentially inflating the scores artificially.

The other term in the MSI metric is the *mask size penalty*. As discussed earlier, a heatmap can highlight more pixels than necessary while still allowing the model to make correct predictions. In such cases, a smaller, more focused mask that conveys the same predictive information is preferable. To penalize unnecessarily large masks, we define the mask size penalty as:

$$\text{MaskPenalty} = \frac{1}{n} \sum_{i=1}^{n} \frac{1}{h \cdot w} \big\| \mathbf{1} \left[ \boldsymbol{M}^i \geq \alpha_{\min} \right] \big\|_1 \tag{5}$$

Here, $\boldsymbol{M}^i \in \mathbb{R}^{h \times w}$ denotes the heatmap for the $i$-th input, and $\mathbf{1}[\cdot]$ is the indicator function that returns 1 for entries satisfying the condition inside the brackets, 0 otherwise. Specifically, this term measures the proportion of pixels with values greater than or equal to the threshold $\alpha_{\min}$, i.e., those considered important. A lower mask penalty indicates a more compact and focused explanation, aligning with the principle of minimality.

The final MSI score combines the base score, which reflects sufficiency and discriminative quality, with the mask penalty, which enforces minimality. The final MSI score is given by:

$$\text{MSI} = \text{BaseScore}_{\text{avg}} - \text{MaskPenalty} \tag{6}$$

Calculating the quality of visual attributions in this way produces scores that better distinguish between valid explanations that differ in parsimony. We refer to the quantitative results in Table 2 for a comparison of MSI against existing fidelity metrics. However, our key contribution in this paper is in developing an explanation generation method that aims to directly maximise this metric, detailed in the following section.

## 3.2 PROPOSED SELF-EXPLAINABLE APPROACH

Building on the minimality-sufficiency logic of the MSI metric, we propose **Learnable Adapter eXplanation (LAX)**, a self-supervised method that learns to generate explanations for a given model without requiring ground-truth annotations. While previous work (Choi et al., 2024; Souibgui et al., 2025) has tackled similar objectives, they typically rely on auxiliary signals that serve as a proxy for ground truth to guide the mask generation process.

To define our objective formally, we draw inspiration from the Information Bottleneck (IB) framework (Tishby & Zaslavsky, 2015). The core idea is to learn a representation that is both *minimal*—containing as little information as possible from the input $\boldsymbol{X}$ as possible—and *sufficient*—preserving the ability to accurately predict the output $\boldsymbol{y}$. Our goal is to learn a mask $\boldsymbol{M}$ such that the masked input $\boldsymbol{T} = \boldsymbol{X} \odot \boldsymbol{M}$ retains only the most relevant information needed for prediction. In our case, this translates to two information-theoretic goals: minimizing the mutual information $I(\boldsymbol{X}, \boldsymbol{T})$ to enforce minimality, and maximizing $I(\boldsymbol{T}, \boldsymbol{y})$ to ensure sufficiency. These can be formulated as a single minimization objective:

$$\mathcal{L} = I(\boldsymbol{X}, \boldsymbol{T}) - \beta \cdot I(\boldsymbol{T}, \boldsymbol{y}),$$

where $\beta$ is a hyperparameter that balances the trade-off between compression and predictive power.

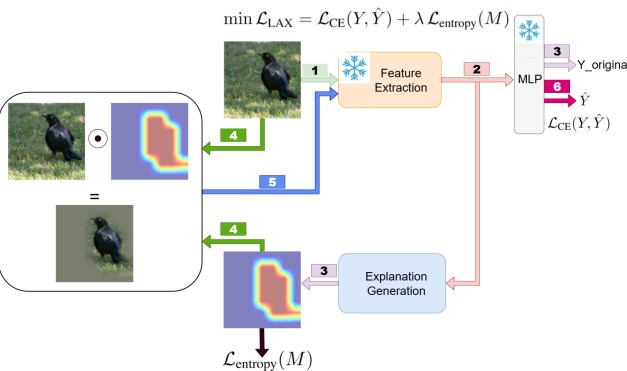

Figure 2: Overview of the proposed LAX framework with respect to a frozen, pre-trained model. **(1)** The image is processed by the feature extractor to obtain feature representations. **(2)** These spatial features are simultaneously sent to the output MLP for classification and to the explanation module. **(3)** The MLP produces the original class prediction, while the explanation module generates a corresponding heatmap. **(4)** The original image is multiplied with the heatmap to produce a masked image. **(5)** The masked image is passed through the frozen feature extractor and MLP. **(6)** The model generates a new prediction based on the masked image.

Figure 2 provides a visual overview of our proposed method. More formally, we describe the specific steps in Algorithm 1. The core idea is to augment any pre-trained black-box model with an explanation module that learns to generate visual explanations as pixel-wise importance heatmaps directly. As a supervision signal, we use the model's accuracy on the input example masked with the proposed heatmap, backpropagated through the explanation module.

Our method operates from a pre-trained base model on the target task. We assume that the base model comprises a feature extraction backbone and a classification head, but otherwise the method is architecture-agnostic. The base model's weights are frozen, and we initialize the new explanation module as a learnable projection from the backbone's feature representations.

The explanation block is modular by design and can adopt various architectures. In our implementation, it is a small Convnet, chosen to efficiently preserve spatial features. This explanation module generates a heatmap in a lower-resolution feature space. Although it is possible to generate a mask at the same resolution as the input image, we empirically found that learning a low-resolution mask and upsampling leads to better and more stable performance.

To ensure interpretability, the heatmap values must lie in the range $[0, 1]$, where a value of 0 indicates that a pixel is unimportant and a value of 1 denotes high importance. To achieve this, we apply a sigmoid activation function to the raw heatmap logits (before upsampling).

---

**Algorithm 1** LAX Framework

---

**Require:** Input image $X$, frozen black-box model $f = f_{\text{MLP}} \circ f_{\text{feat}}$, learnable explanation module $g$

**Ensure:** Mask $M$, masked image $T$, prediction $\hat{y}$

1: **Step 1:** Extract features: $F \leftarrow f_{\text{feat}}(X)$
2: **Step 2:**
      (a) Predict class: $\hat{y}_{\text{orig}} \leftarrow f_{\text{MLP}}(F)$
      (b) Generate heatmap: $M \leftarrow g(F)$
3: **Step 3:** Upsample heatmap $M$ to match input resolution (if needed)
4: **Step 4:** Compute masked image: $T \leftarrow X \odot M$
5: **Step 5:** Predict from masked image: $\hat{y} \leftarrow f_{\text{MLP}}(f_{\text{feat}}(T))$
6: **return** Explanation heatmap $M$, masked image $T$, prediction $\hat{y}$

---

As previously introduced, the IB-based loss involves two mutual information terms: (1) $I(X, T)$, which we minimize to enforce minimality, and (2) $I(T, y)$, which we maximize to ensure sufficiency. Optimizing these terms directly is generally intractable; therefore, we propose tractable approximations for each term.

Table 2: Qualitative and quantitative comparison between LAX and Grad-CAM on randomly chosen examples from different datasets. In the column 'Visual', the the first image is the original input, the second shows the mask generated by LAX, and the third displays the mask produced by Grad-CAM. The threshold $\alpha_{\min}$ is 0.5 for MNIST and CUB-200, and 0.4 for CIFAR-10.

| Visual | Insertion (Pixel Value) (Morf) ↑ | | Deletion (Pixel Value) (Morf) ↓ | | Insertion (%) (Morf) ↑ | | Deletion (%) (Morf) ↓ | | Fid- (Insertion) (Pixel Value) ↓ | | Fid+ (Deletion) (Pixel Value) ↑ | | Fid- (Insertion) (%) ↓ | | Fid+ (Deletion) (%) ↑ | | Base Score ↑ | | Mask Penalty ↓ | | MSI Score ↑ | |
|---|---|---|---|---|---|---|---|---|---|---|---|---|---|---|---|---|---|---|---|---|---|---|
| | LAX | Grad CAM | LAX | Grad CAM | LAX | Grad CAM | LAX | Grad CAM | LAX | Grad CAM | LAX | Grad CAM | LAX | Grad CAM | LAX | Grad CAM | LAX | Grad CAM | LAX | Grad CAM | LAX | Grad CAM |
|  | **0.745** | 0.605 | **0.125** | 0.175 | **0.91** | 0.53 | **0.07** | 0.15 | **0.275** | 0.393 | **0.886** | 0.818 | **0.106** | 0.487 | **0.938** | 0.847 | **0.68** | -0.015 | **0.185** | 0.553 | **0.495** | -0.568 |
|  | **0.965** | 0.565 | 0.615 | **0.585** | **0.89** | 0.71 | **0.39** | 0.45 | **0.303** | 0.498 | 0.4799 | **0.497** | **0.256** | 0.375 | **0.607** | 0.598 | **0.46** | -0.11 | **0.351** | 0.37 | **0.109** | -0.48 |
|  | **0.965** | 0.875 | **0.075** | 0.255 | **0.93** | 0.97 | 0.09 | **0.07** | **0.008** | 0.086 | **0.802** | 0.676 | 0.022 | **0.006** | 0.772 | **0.817** | **0.66** | 0.54 | **0.204** | 0.218 | **0.456** | 0.322 |
|  | **0.955** | 0.765 | **0.565** | 0.255 | 0.75 | **0.77** | 0.31 | **0.25** | **0.115** | 0.249 | 0.488 | **0.715** | 0.304 | **0.243** | 0.675 | **0.722** | **0.943** | 0.608 | **0.318** | 0.73 | **0.625** | -0.122 |

**Minimizing $I(X, T)$ via entropy regularization.** We define $T = X \odot M$, where $M$ is the learnable mask and $X$ is the fixed input. Since $X$ remains constant, the only way to reduce the mutual information $I(X, T)$ is by limiting the information retained in $T$, effectively encouraging the mask $M$ to be sparse. A common approach is to apply an $\ell_1$ regularization on $M$, which promotes sparsity. While this method is effective to some extent, it often produces masks that are less sharp and harder to interpret. In contrast, we opt to use an entropy-based regularization, which yields sharper and more semantically meaningful masks that concentrate more on important regions.

Our entropy-based loss is formulated as follows:

$$\mathcal{L}_{\text{entropy}} = \frac{1}{n} \sum_{i=1}^{n} \left[ -\sum_j P_{i,j} \log(P_{i,j} + \varepsilon) \right], \tag{7}$$

where

$$P_i = \text{softmax}\left( \frac{\max(0, \boldsymbol{M}_i)}{t} \right),$$

$\boldsymbol{M}_i \in \mathbb{R}^{h \times w}$ is the mask for sample $i$,

$t$ is a temperature parameter controlling the sharpness,

$\varepsilon$ is a small constant for numerical stability,

$n$ is the batch size.

Minimizing this entropy loss results in heatmaps that assign high confidence primarily to the most critical regions, effectively enforcing minimality.

**Maximizing $I(\boldsymbol{T}, \boldsymbol{y})$ via cross-entropy approximation.** To ensure that the masked input $\boldsymbol{T} = \boldsymbol{X} \odot \boldsymbol{M}$ retains sufficient information for accurate prediction, we maximize the mutual information $I(\boldsymbol{T}, \boldsymbol{y})$. Using the approximation introduced by Amjad & Geiger (2019), we have:

$$\max I(\boldsymbol{T}; \boldsymbol{y}) = \underbrace{H(\boldsymbol{y})}_{\text{constant}} - H(\boldsymbol{y} \mid \boldsymbol{T})$$

$$\iff \quad \min H(\boldsymbol{y} \mid \boldsymbol{T}) \approx \mathcal{L}_{\text{CE}}(\boldsymbol{y}, \hat{\boldsymbol{y}}) \tag{8}$$

where $H(\boldsymbol{y})$ and $H(\boldsymbol{y} \mid \boldsymbol{T})$ represent the marginal and conditional entropies of $\boldsymbol{y}$, respectively. Here, $\boldsymbol{y}$ denotes the ground-truth label, and $\hat{\boldsymbol{y}}$ denotes the model's prediction on the masked input. Note that $H(\boldsymbol{y})$ is constant because the label distribution is fixed.

**Final objective.** Combining both terms, the LAX module final loss is:

$$\min \mathcal{L}_{\text{LAX}} = \mathcal{L}_{\text{CE}}(\boldsymbol{y}, \hat{\boldsymbol{y}}) + \lambda \, \mathcal{L}_{\text{entropy}}(\boldsymbol{M}), \tag{9}$$

where $\lambda$ is a hyperparameter balancing prediction accuracy and mask sparsity. Minimizing this objective enables the explanation module to learn compact, informative, and interpretable masks without reliance on ground-truth explanations.

Table 3: Quantitative results on the Synthetic MNIST dataset. LAX outperforms all baseline methods across all standard metrics and the proposed MSI metric, indicating strong minimality and sufficiency under low-ambiguity settings.

| Method | Insertion (Pixel Value) ↑ (Morf) | Deletion (Pixel Value) ↓ (Morf) | Insertion (%) ↑ (Morf) | Deletion (%) ↓ (Morf) | Fid- (Insertion) ↓ (Pixel Value) | Fid+ (Deletion) ↑ (Pixel Value) | Fid- (Insertion) ↓ (%) | Fid+ (Deletion) ↑ (%) | Base Score ↑ ($\alpha_{\min} = 0.5$) | Mask Penalty ↓ | MSI Score ↑ |
|---|---|---|---|---|---|---|---|---|---|---|---|
| Grad-CAM | 0.744 | 0.250 | 0.816 | 0.198 | 0.255 | 0.745 | 0.183 | 0.798 | 0.509 | 0.418 | 0.091 |
| Grad-CAM++ | 0.712 | 0.268 | 0.804 | 0.204 | 0.287 | 0.728 | 0.194 | 0.792 | 0.456 | 0.371 | 0.085 |
| Layer-CAM | 0.716 | 0.266 | 0.805 | 0.203 | 0.282 | 0.73 | 0.194 | 0.793 | 0.462 | 0.378 | 0.084 |
| KPCA-CAM | 0.612 | 0.329 | 0.825 | 0.201 | 0.388 | 0.666 | 0.175 | 0.794 | 0.316 | 0.187 | 0.128 |
| Finer-CAM | 0.616 | 0.347 | 0.742 | 0.244 | 0.382 | 0.649 | 0.256 | 0.751 | 0.301 | 0.318 | -0.027 |
| **LAX** | **0.917** | **0.142** | **0.906** | **0.162** | **0.085** | **0.851** | **0.092** | **0.833** | **0.787** | **0.205** | **0.582** |

## 4 EXPERIMENTS AND RESULTS

**Experimental Setup.** To evaluate our proposed method, we compare it against several existing baseline techniques for explanation generation, using both existing metrics and our proposed MSI metric. For MSI, we report three separate components: the base score, the mask penalty, and the final MSI score. Each heatmap is processed according to the requirements of the respective evaluation metric (e.g., insertion, deletion, Fid, MSI) to compute the corresponding score.

**Baseline Methods.** We compare our LAX method against several state-of-the-art techniques for generating explanation heatmaps: Grad-CAM (Selvaraju et al., 2017), Grad-CAM++ (Chattopad-hay et al., 2018), Layer-CAM (Jiang et al., 2021), KPCA-CAM (Karmani et al., 2024), and Finer-CAM (Zhang et al., 2025).

**Model.** All experiments use a ResNet18 architecture trained for 500 epochs using standard cross-entropy loss. After the base model is trained on the classification task, we initialize and train the LAX module for another 500 epochs. During this phase, we optimize a combination of the classification loss computed on the masked input and an entropy-based regularization loss. Additionally, we monitor the average value of the generated masks, which helps guide hyperparameter tuning, particularly the selection of the weighting coefficient $\lambda$ for the entropy loss.

**Baseline Metrics.** To assess the quality of different explanation methods, we employ several widely used evaluation metrics in addition to our proposed MSI score. These include **Insertion** and **Deletion** (Petsiuk, 2018), as well as **Fid-Insertion** and **Fid-Deletion** (Zheng et al., 2025). Each metric evaluates the model's response as information is either progressively added to or removed from the input based on the explanation heatmap. We report two variants for each metric: Pixel-based and percentage-based. More details about the baseline metrics and their interpretation are available as supplementary material.

**Datasets.** We evaluate our method across three datasets selected to reflect varying levels of complexity. In the simplest case, we use an augmented variant of MNIST (Deng, 2012) which randomly repositions target digits and adds background noise to make explanation generation non-trivial. We also use CIFAR-10 (Krizhevsky et al., 2009) as a standard evaluation dataset, and CUB-200 (Welinder et al., 2010) as a more realistic, higher-resolution setting.

**Metric Evaluation.** Table 2 shows the behavior of the various metrics on heatmaps produced by Grad-CAM and by our method (LAX). Notably, MSI is the most consistent and amplifies differences when explanations differ substantially in quality. For example, on the first sample, LAX outperforms Grad-CAM by over 1.0 point in MSI, whereas Insertion and Deletion differ by only 0.15. This indicates that MSI is more sensitive to qualitative differences, rewarding explanations that are both sufficient and compact. Furthermore, MSI values for Grad-CAM remain negative or close to zero across several examples, aligning with visual impressions of noisy or diffuse saliency maps, while LAX consistently yields positive MSI scores that reflect minimal and sufficient regions.

**Quantitative Method Evaluation.** The quantitative results of the different explainability methods are summarized in Table 3, Table 4, and Table 5, corresponding to the datasets Synthetic MNIST, CUB-200, and CIFAR-10, respectively. See Figures 3 and 4 for qualitative comparisons.

LAX consistently outperforms baselines on Synthetic MNIST across all metrics, achieving higher MSI scores due to minimal mask size penalties. On CUB-200, traditional metrics are inconclusive in preferring Grad-CAM, Finer-CAM or LAX, but LAX attains a higher MSI score despite similar mask size, showing more discriminative focus.

On CIFAR-10, LAX leads in Insertion-based metrics while Grad-CAM shows better Deletion scores. This divide indicates the presence of multiple informative regions, which is more readily discriminated by the MSI metric. Across all datasets, the MSI metric proves valuable for distinguishing between methods that appear similar under conventional metrics.

| Method | Insertion (Pixel Value) (Morf) ↑ | Deletion (Pixel Value) (Morf) ↓ | Insertion (%) (Morf) ↑ | Deletion (%) (Morf) ↓ | Fid- (Insertion) (Pixel Value) ↓ | Fid+ (Deletion) (Pixel Value) ↑ | Fid- (Insertion) (%) ↓ | Fid+ (Deletion) (%) ↑ | Base Score ($\alpha_{\min}=0.5$) ↑ | Mask Penalty ↓ | MSI Score ↑ |
|---|---|---|---|---|---|---|---|---|---|---|---|
| Grad-CAM | 0.665 | **0.412** | 0.768 | **0.284** | 0.198 | 0.421 | 0.103 | **0.521** | 0.244 | 0.264 | -0.019 |
| Grad-CAM++ | 0.654 | 0.431 | 0.763 | 0.291 | 0.21 | 0.411 | 0.109 | 0.517 | 0.23 | 0.256 | -0.026 |
| Layer-CAM | 0.662 | 0.414 | 0.763 | 0.291 | 0.203 | 0.422 | 0.112 | 0.517 | 0.243 | 0.275 | -0.031 |
| KPCA-CAM | 0.565 | 0.556 | 0.757 | 0.318 | 0.294 | 0.315 | 0.114 | 0.505 | 0.058 | **0.148** | -0.091 |
| Finer-CAM | 0.646 | 0.455 | **0.774** | 0.295 | 0.218 | 0.383 | **0.099** | 0.504 | 0.197 | 0.221 | -0.023 |
| LAX | **0.767** | 0.415 | 0.772 | 0.317 | **0.137** | **0.426** | 0.101 | 0.491 | **0.445** | 0.265 | **0.18** |

Table 4: Quantitative results on the CUB-200 (Birds) dataset. While several methods perform similarly on standard metrics, LAX achieves the highest MSI score, showing success in optimizing for this metric.

| Method | Insertion (Pixel Value) (Morf) ↑ | Deletion (Pixel Value) (Morf) ↓ | Insertion (%) (Morf) ↑ | Deletion (%) (Morf) ↓ | Fid- (Insertion) (Pixel Value) ↓ | Fid+ (Deletion) (Pixel Value) ↑ | Fid- (Insertion) (%) ↓ | Fid+ (Deletion) (%) ↑ | Base Score ($\alpha_{\min}=0.4$) ↑ | Mask Penalty ↓ | MSI Score ↑ |
|---|---|---|---|---|---|---|---|---|---|---|---|
| Grad-CAM | 0.637 | **0.384** | 0.668 | **0.339** | 0.301 | **0.55** | 0.272 | **0.593** | **0.374** | 0.599 | -0.225 |
| Grad-CAM++ | 0.62 | 0.395 | 0.662 | 0.342 | 0.317 | 0.541 | 0.278 | 0.59 | 0.353 | 0.576 | -0.222 |
| Layer-CAM | 0.626 | 0.387 | 0.659 | 0.341 | 0.311 | 0.547 | 0.281 | 0.591 | 0.364 | 0.596 | -0.232 |
| KPCA-CAM | 0.371 | 0.674 | 0.596 | 0.447 | 0.566 | 0.268 | 0.343 | 0.489 | -0.094 | **0.272** | -0.367 |
| Finer-CAM | 0.621 | 0.411 | 0.674 | 0.352 | 0.316 | 0.525 | 0.266 | 0.581 | 0.347 | 0.553 | -0.206 |
| LAX | **0.739** | 0.581 | **0.735** | 0.423 | **0.207** | 0.364 | **0.201** | 0.516 | 0.345 | 0.337 | **0.007** |

Table 5: Quantitative results on the CIFAR-10 dataset. MSI is able to rate a method highly even when multiple plausible explanations exist.

## 5 CONCLUSION

In this paper, we introduced Minimality-Sufficiency Integration (MSI), a novel metric for quantifying the quality of visual explanations without relying on ground-truth saliency annotations. MSI addresses key limitations of existing fidelity-based metrics by jointly evaluating the sufficiency and minimality of explanations, while tolerating multiple valid explanations. Our experiments show that MSI offers more consistent and interpretable scores across varying tasks and datasets.

To complement this metric, we proposed LAX (Learnable Adapter eXplanation), an explanation module that learns to generate compact and informative saliency maps by directly optimizing toward the MSI objective. LAX operates as a lightweight adapter over frozen black-box models and requires no ground-truth explanations, making it broadly applicable. Quantitative and qualitative evaluations across three diverse datasets confirm that LAX outperforms established baselines on various metrics, including MSI. Future work will explore extending both MSI and LAX to multi-modal data and large transformer models.

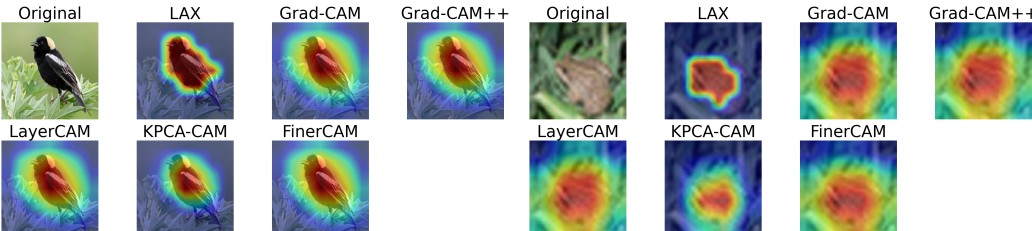

Figure 3: Qualitative example on CUB-200.    Figure 4: Qualitative example on CIFAR-10.

ACKNOWLEDGMENTS

This work was funded by the European Union - by Next Generation EU, under Grant No. SDC007/25/000075; and by ELSA – European Lighthouse on Secure and Safe AI under Grant Agreement No. 101070617.

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

# A APPENDIX

## A.1 METRICS

### A.1.1 MSI METRIC

**Interpreting the MSI Score.** To interpret the MSI score meaningfully, it is important to consider both its theoretical and practical range. The MSI score ranges theoretically from $-2$ to $+1$. A score of $-2$ reflects an unrealistic scenario in which the model achieves perfect prediction when the entire image is masked out (i.e., no visual information is shown), while achieving zero accuracy when the full original image is provided. As such, we focus on the practical range of MSI scores, which typically lies between $-1$ and $+1$.

The score is computed with respect to a chosen threshold $\alpha_{\min}$, which determines the cutoff for important pixels. Interpretation of the score at a given $\alpha_{\min}$ is as follows:

- **Score close to $+1$:** Indicates a high-quality explanation. The mask captures minimal yet sufficient information to make correct predictions, and the region above $\alpha_{\min}$ likely includes all relevant evidence (i.e., one or more complete solutions).

- **Score close to $-1$:** Suggests a poor mask. The retained region fails to provide sufficient information for prediction, indicating that the explanation is not capturing meaningful or discriminative features.

- **Score near $0$:** Can indicate one of two cases:
    - The mask is too large, including many unnecessary pixels. In such cases, qualitative inspection may help determine whether a large mask is truly required (e.g., in tasks with large or diffuse objects), or whether the explanation lacks focus.
    - The mask is minimal and predictive, but other valid solutions exist in the input that are not captured above the threshold $\alpha_{\min}$. This situation arises in settings where multiple plausible explanations can lead to correct predictions.

Overall, MSI provides a more nuanced evaluation than standard metrics by jointly accounting for sufficiency, minimality, and the possibility of multiple valid solutions. This makes it particularly valuable for tasks involving complex inputs, multi-modal reasoning, or interpretability under uncertainty. While the goal is to achieve an MSI score closer to $+1$, a score near $0$ does not necessarily indicate a poor explanation. In some cases, it may reflect either the presence of alternative explanations or task-specific requirements for larger masks.

In addition, a higher value of $\alpha_{\min}$ indicates that the mask is highly effective at separating important from unimportant pixels. This means the model can rely on a smaller, more confidently identified region to make accurate predictions, reflecting a more precise and discriminative explanation.

By default, MSI is computed using classification accuracy as the predictive score, but it is easily extendable to other measures such as model confidence scores, allowing flexible application across different interpretability contexts.

### A.1.2 OTHER METRICS

To assess the quality of different explanation methods, we employ several widely used evaluation metrics in addition to our proposed MSI score. These include **Insertion** and **Deletion** Petsiuk (2018), as well as **Fid-Insertion** and **Fid-Deletion** Zheng et al. (2025). Each metric evaluates the model's response as information is either progressively added to or removed from the input based on the explanation heatmap. We report two variants for each metric: Pixel-based and percentage-based.

- **(Pixel value-based)**: In this setting, pixels are inserted or deleted by thresholding the heatmap values directly. For example, we may iteratively remove or restore pixels above a threshold value—e.g., 0.98, 0.96, and so on—based on heatmap intensity.

- **(Percentage-based)**: Here, pixels are sorted by heatmap values, and the top $k\%$ are inserted or deleted in increasing steps (e.g., top 2%, 4%, etc.).

The metrics are interpreted as follows:

- **Insertion**: Measures how quickly the model's accuracy increases as pixels are progressively inserted. *Higher is better.*

- **Deletion**: Measures how quickly the model's performance degrades as pixels are progressively removed. *Lower is better.*

- **Fid-Insertion (Fidelity-Insertion)**: Measures the change in the model's output when comparing the full image to a progressively inserted version based on the explanation heatmap. *Lower is better.*

- **Fid-Deletion (Fidelity-Deletion)**: Measures the change in the model's output when comparing the full image to a progressively deleted version based on the heatmap. *Higher is better.*

This comprehensive set of metrics—covering prediction confidence, information contribution, and fidelity—enables us to thoroughly evaluate the effectiveness and faithfulness of different explanation methods.

## A.2 DATASETS

**1. Synthetic MNIST.** The first dataset is a synthetic extension of the MNIST digit dataset LeCun et al. (2010). We designed this dataset to increase the visual complexity of MNIST and better test the explanatory power of heatmaps. Specifically, we generate background images using random pixel values, embed randomly placed geometric shapes (e.g., boxes), and insert randomly scaled and positioned MNIST digits. This setup introduces distractors while maintaining the digit as the key predictive region. A key advantage of this dataset is that, in most cases, only the digit itself is required for accurate prediction—making it ideal for validating the minimality objective in our MSI metric. In practice, we observe that MSI scores on this dataset tend to be close to $+1$, reflecting the presence of a single dominant solution.

**2. Caltech-UCSD Birds-200-2011 (CUB-200).** The second dataset is the Caltech-UCSD Birds-200-2011 dataset (CUB-200-2011) Wah et al. (2011), which contains 200 fine-grained bird species. For our experiments, we randomly select 20 classes. This dataset presents a more challenging scenario: while all birds share certain visual features (e.g., wings, beaks), each class requires the model to attend to specific discriminative cues for accurate classification. Although most examples likely have a single dominant region of interest, a small number of instances may contain multiple disjoint yet sufficient regions. Consequently, on this dataset, we expect the MSI score to be significantly higher than 0.

**3. CIFAR-10.** The third dataset is CIFAR-10 Krizhevsky (2009), which contains natural images from 10 object categories. Based on manual inspection and qualitative examples (e.g., as shown in Figures 17, 18, 19, and 20), we observe that many images contain multiple disjoint regions that can independently support accurate predictions. This characteristic makes CIFAR-10 a representative case for evaluating explanation methods under multi-solution scenarios. Accordingly, we expect the MSI scores on this dataset to be lower and closer to zero—which, in this context, is not an indication of poor mask quality, but rather reflects the inherent ambiguity in what constitutes a sufficient explanation.

## A.3 IMPLEMENTATION DETAILS

### A.3.1 EXPLANATION GENERATION

To generate explanations, we use a lightweight convolutional module that operates on the feature representations extracted from the base model. In our setup, we use a lightweight ResNet as the base model, and the output of its last convolutional layer is used as the input to the explanation generation block.

Since the three datasets have different input image sizes, we modify certain internal parameters of the base model—such as stride and padding—to ensure effective training. As a result of these adjustments, the output feature map shapes vary across datasets. Specifically:

- For **Synthetic MNIST**, the feature map shape is $8 \times 8 \times 512$.

- For **CUB-200**, the feature map shape is $7 \times 7 \times 512$.
- For **CIFAR-10**, the feature map shape is $4 \times 4 \times 512$.

In both the Synthetic MNIST and CUB-200 cases, we do not apply upsampling, and the explanation heatmap is generated at the native resolution of the feature map. However, in the case of CIFAR-10, we apply upsampling within the explanation module to obtain an $8 \times 8$ heatmap for better spatial coverage and interpretability.

The explanation block progressively reduces the channel dimension from the initial 512 channels to a single-channel output representing the heatmap. This is achieved through a series of convolutional layers:

$$512 \rightarrow 256 \rightarrow 128 \rightarrow 64 \rightarrow 1$$

This final single-channel output corresponds to the explanation heatmap, with spatial resolution either $8 \times 8$ (for Synthetic MNIST and CIFAR-10) or $7 \times 7$ (for CUB-200). These resolutions are not fixed by design but were found empirically to perform well in practice. Lower spatial resolutions encourage the network to focus on salient, coarse-level regions and tend to yield more interpretable and stable explanations. After generating the low-resolution heatmap, we upsample it to the original input image size using bilinear interpolation to enable pixel-wise alignment and visualization.

**Hyperparameters for Explanation Generation** The explanation generation module was trained using different hyperparameter settings tailored to each dataset to ensure optimal learning behavior. Table 6 summarizes the learning rate, optimizer, entropy regularization weight ($\lambda_{\text{entropy}}$), and temperature parameter ($T$) used during training. These values were selected based on empirical validation performance and qualitative stability of the generated explanations.

Table 6: Hyperparameters for Explanation Generation across datasets.

| Hyperparameter | Synthetic MNIST | CUB 200 | CIFAR 10 |
|---|---|---|---|
| Learning Rate | 0.001 | 0.0001 | 0.001 |
| Optimizer | Adam | Adam | Adam |
| $\lambda_{\text{entropy}}$ | 5 | 5 | 2 |
| Temperature ($t$) | 0.5 | 0.5 | 0.5 |

## A.4 ADDITIONAL RESULTS

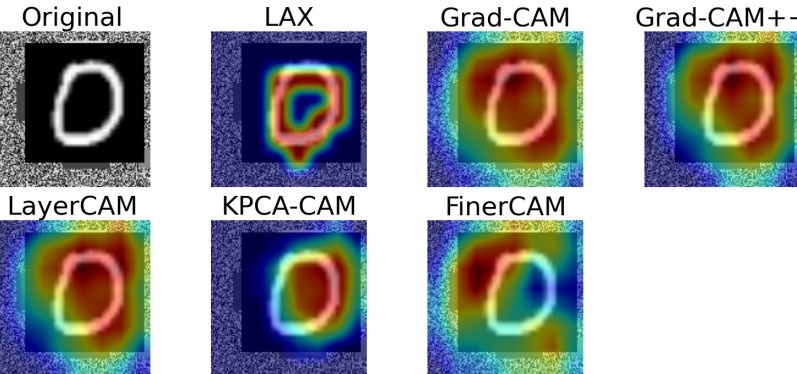

Figure 5: Qualitative example on Synthetic MNIST.

## A.5 NOTE ON LLM USAGE

Large language models were used in a limited capacity during the drafting of this paper, to improve clarity and polish the use of language. All technical contributions, ideas and content are solely attributable to the human authors.

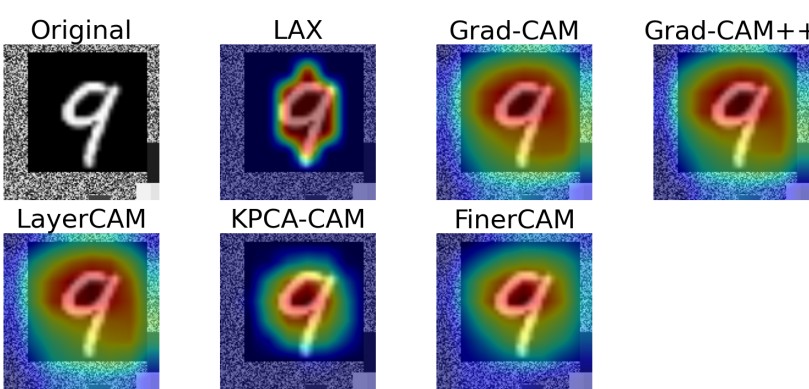

Figure 6: Qualitative example on Synthetic MNIST.

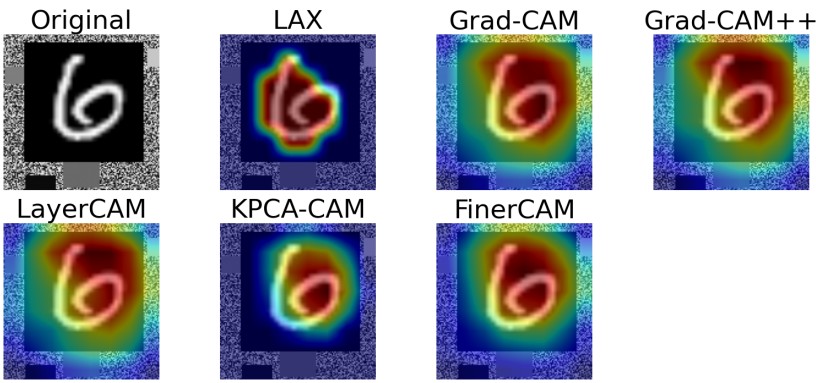

Figure 7: Qualitative example on Synthetic MNIST.

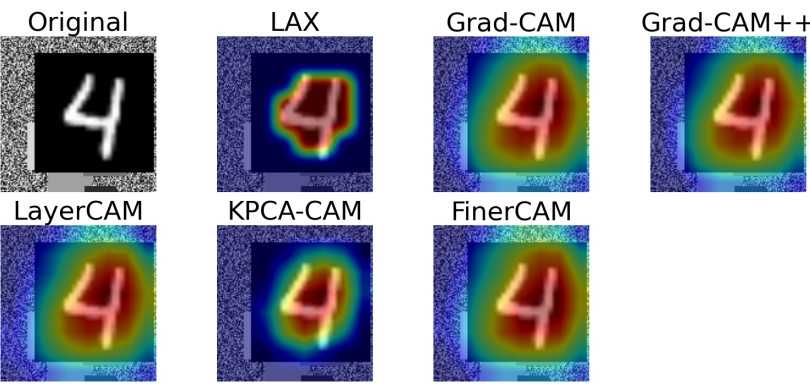

Figure 8: Qualitative example on Synthetic MNIST.

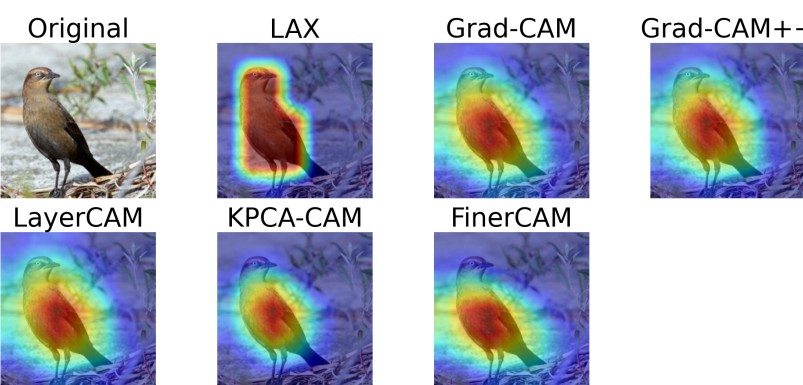

Figure 9: Qualitative example on CUB-200.

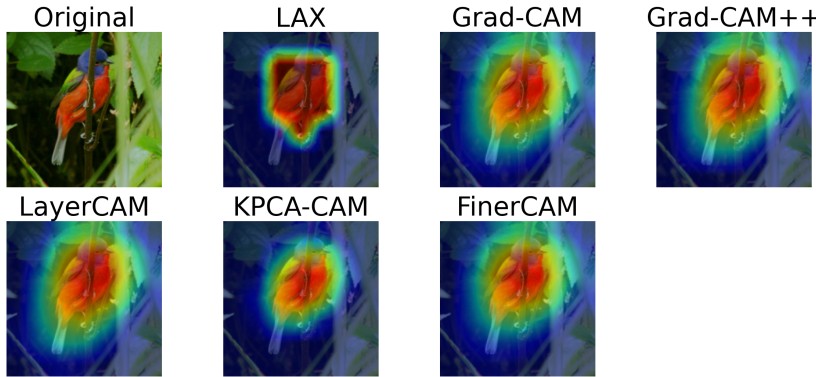

Figure 10: Qualitative example on CUB-200.

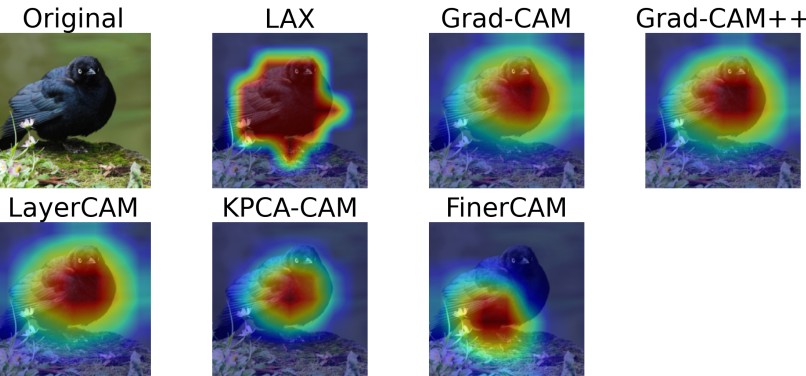

Figure 11: Qualitative example on CUB-200.

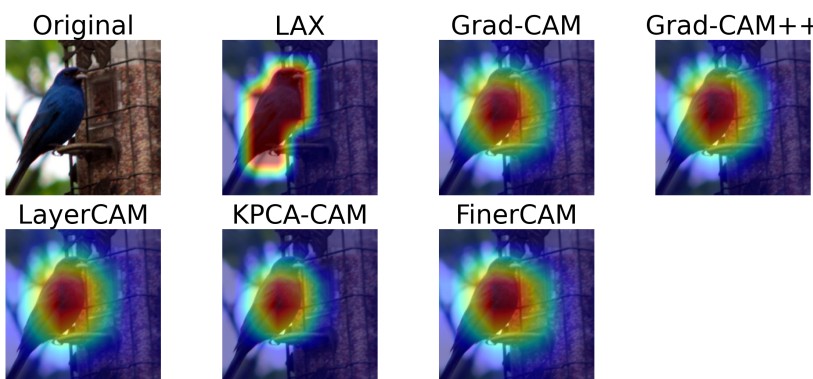

Figure 12: Qualitative example on CUB-200.

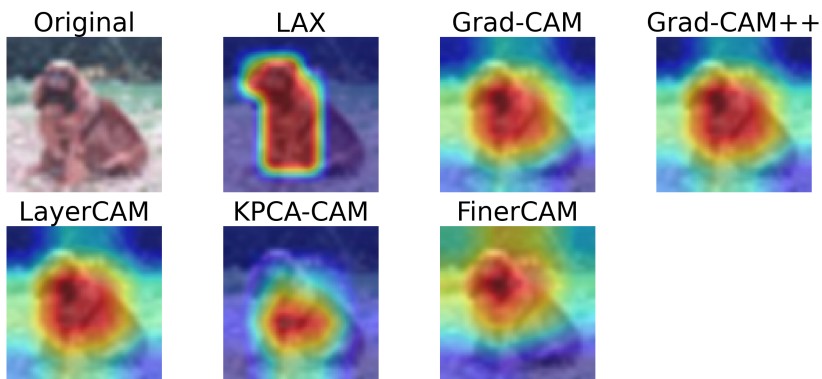

Figure 13: Qualitative example on CIFAR-10.

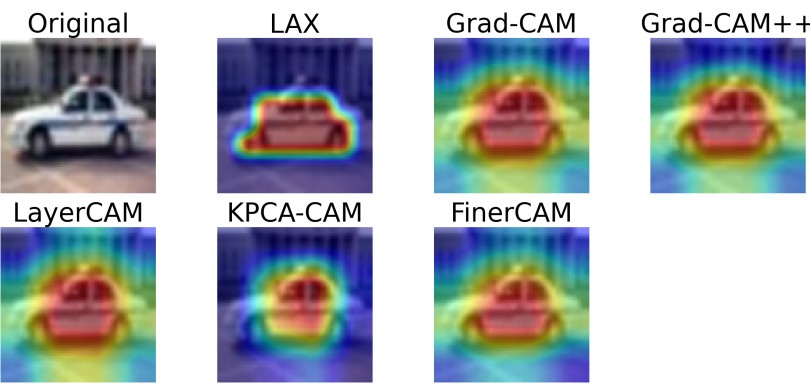

Figure 14: Qualitative example on CIFAR-10.

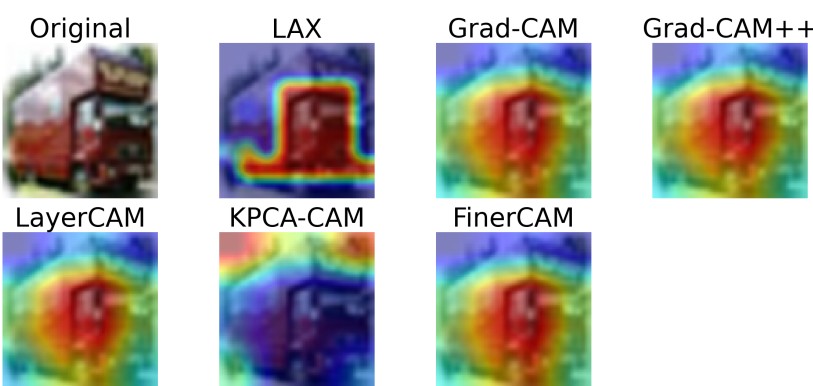

Figure 15: Qualitative example on CIFAR-10.

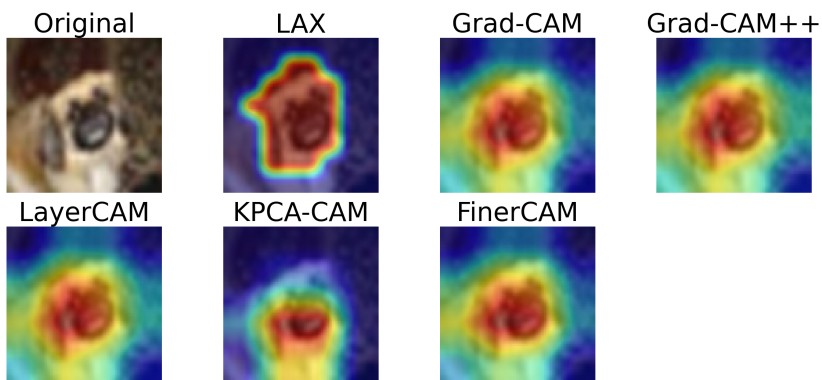

Figure 16: Qualitative example on CIFAR-10.

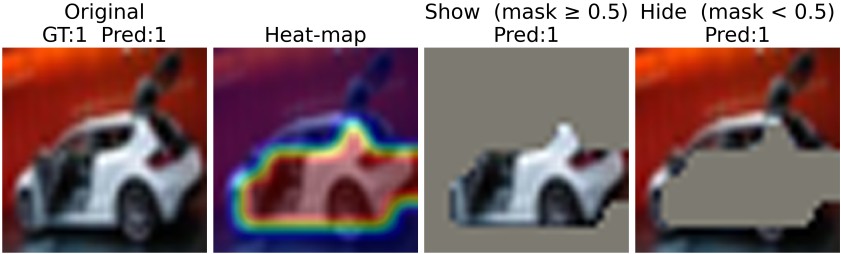

Figure 17: Example illustrating multiple valid solutions. From left to right: original image, full mask, thresholded mask region (values $\geq 0.5$), and its complement (values $< 0.5$).

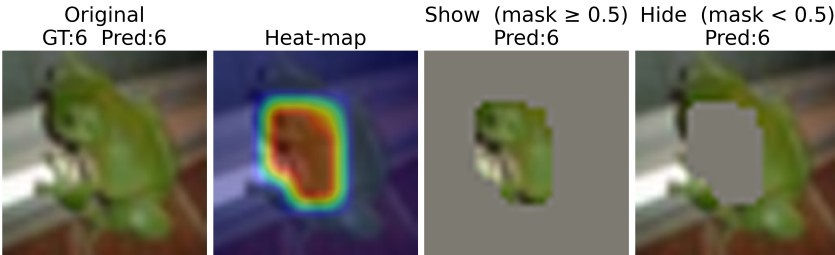

Figure 18: Example illustrating multiple valid solutions. From left to right: original image, full mask, thresholded mask region (values $\geq 0.5$), and its complement (values $< 0.5$).

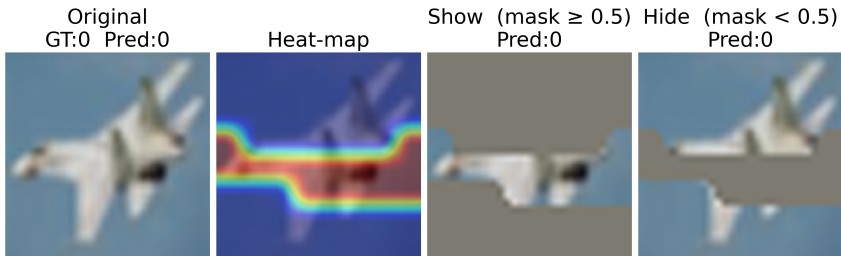

Figure 19: Example illustrating multiple valid solutions. From left to right: original image, full mask, thresholded mask region (values $\geq 0.5$), and its complement (values $< 0.5$).

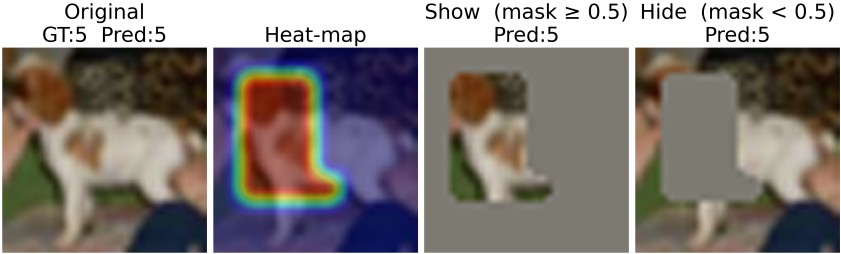

Figure 20: Example illustrating multiple valid solutions. From left to right: original image, full mask, thresholded mask region (values $\geq 0.5$), and its complement (values $< 0.5$).

