# OpenReview forum: "Learning Quantifiable Visual Explanations Without Ground-Truth"
_ICLR.cc/2026/Conference — Submitted to ICLR 2026_

### Official Review · Reviewer_Gsqk · 2025-10-27

**Soundness:** 3
**Presentation:** 2
**Contribution:** 2
**Rating:** 4
**Confidence:** 5

**Summary:**

This paper introduces Minimality-Sufficiency Integration (MSI), a new metric for evaluating visual explanations without ground-truth saliency annotations, and Learnable Adapter eXplanation (LAX), a self-supervised module trained to generate compact, sufficient saliency maps that optimize MSI directly.
MSI integrates sufficiency (the explanation retains information needed for correct prediction) and minimality (it avoids redundant regions), and tolerates multiple valid explanations.
LAX, inspired by the Information Bottleneck framework, learns an explanation mask on top of a frozen backbone without altering predictive performance.
Experiments on Synthetic-MNIST, CUB-200, and CIFAR-10 show that LAX outperforms Grad-CAM variants on both traditional fidelity metrics and the proposed MSI score.

**Strengths:**

- The paper directly tackles a major challenge in explainability (evaluating without ground truth) by introducing a metric grounded in sufficiency and minimality.

- The combination of a differentiable evaluation metric (MSI) and a compatible self-supervised explanation module (LAX) is conceptually elegant and generally applicable to pretrained models.

- The paper is easy to follow, with thorough tables and visualizations demonstrating improved localization and focus over CAM-based baselines.

**Weaknesses:**

- The paper only evaluates on image classification and only against CAM-style methods (Grad-CAM, Grad-CAM++, Layer-CAM, etc.). This omits important region-based attribution work such as XRAI (Kapishnikov et al., ICCV 2019), which also emphasizes compact, region-level explanations and introduces similar perturbation-based evaluation curves (AIC/SIC). The lack of direct comparison or discussion of XRAI weakens claims of novelty.

- MSI’s integration of minimality and sufficiency is a thoughtful extension, but conceptually similar to XRAI’s “region-based sufficiency curves” and fidelity metrics. The theoretical innovation is modest, and the proposed IB connection remains heuristic.

- LAX is trained to optimize MSI, and performance is then evaluated primarily with MSI. Without external or human evaluation, it’s unclear whether the explanations are genuinely better or merely optimized for the metric.

- No experiments beyond vision classification, no transformer-based or multimodal examples, and no human-alignment studies. This limits the claim that MSI captures intuitive explanation quality.

**Questions:**

- How does MSI relate quantitatively to XRAI’s AIC/SIC metrics?

- Would LAX or MSI generalize to non-visual or transformer-based models?

- Does MSI correlate with human perception of explanation quality or just model fidelity?

- How does MSI behave under adversarial perturbations or for explanations of incorrect predictions?

- How sensitive is MSI to the choice of α_min and integration bounds?

- Would human raters’ preferences correlate with MSI scores?

- How stable is the LAX module across different random seeds or model checkpoints?

---

### Official Review · Reviewer_gg79 · 2025-10-31

**Soundness:** 2
**Presentation:** 2
**Contribution:** 2
**Rating:** 2
**Confidence:** 4

**Summary:**

The paper proposes Minimality-Sufficiency Integration (MSI), a perturbation-based evaluation metric for saliency or relevance maps. MSI aims to unify some existing faithfulness metrics, combining different aspects of previous perturbation metrics such as prediction performance comparison between high and low importance regions, thresholded MoRF-Insertion and Deletion, and a penalty term for less-sparse explanations. The paper also introduces Learnable Adapter eXplanation (LAX), an auxiliary “explanation network” trained on top of a frozen classifier to generate saliency maps that optimize MSI scores. They evaluate the LAX method on Synthetic MNIST, CIFAR-10, and a CUB-200 subset, comparing LAX to Grad-CAM variants on a ResNet-18 architecture.

**Strengths:**

- **Important Topic**: The evaluation of explanation methods in quantitative as well as theoretical terms is an important topic and helps improve trust in interpretability methods.
- **Reasonable Goal**: Combining sufficiency and minimality in a single metric is a reasonable goal for a metric and addresses some shortcomings in existing metrics.
- **Simple Metric Implementation**: The MSI appears to be simple to implement.

**Weaknesses:**

- **Weak motivation and limited validation of MSI**: The paper does not convincingly demonstrate that MSI better captures attribution quality than existing metrics. MSI is essentially a composition of prior perturbation metrics with an additional sparsity penalty and a tunable selection threshold. To highlight its usefulness, the paper would need to provide more extensive empirical validation across multiple models, attribution methods, datasets, and hyperparameter regimes. The presented quantitative evidence (Table 2) is based on only four samples from a single ResNet-18 model using Grad-CAM and LAX, which is insufficient to draw general conclusions. Methods such as DeepLIFT, LRP or LIME are cited but never tested. The dependence of MSI on the hyperparameter alpha_min (the importance threshold) is especially underexplored. This parameter directly affects which regions are considered relevant and can behave inconsistently across attribution methods with different sparsity levels.
- **Insufficient LAX Experiments**: The LAX adapter is evaluated only on Grad-CAM variants with a single ResNet-18 architecture. The hyperparameter alpha_min changes across datasets without justification, and it is unclear whether any train/test split is used. As a result, it is not possible to assess whether the explanations generalize beyond the training data. The reported results are insufficient to draw meaningful conclusions about the proposed method.
- **Missing discussion of LAX Limitations**: Conceptual limitations of the LAX module are not addressed sufficiently. LAX introduces another model that learns to produce explanations for a frozen model optimizing for targets that are related to maximizing their introduced MSI metric. The paper does not analyze whether these explanations remain faithful or robust, especially for off-manifold or out-of-distribution inputs. Without such analysis, it is unclear whether LAX produces genuinely interpretable or merely overfitted saliency maps.
- **Missing computational analysis**: The paper provides no discussion of the computational cost of computing MSI or training LAX. Without runtime or complexity analysis, it is difficult to assess whether the proposed method is practical compared to existing explanation methods.

**Questions:**

- **Sensitivity of alpha_min**: How sensitive is MSI to the choice of this hyperparameter? Why is it varied across datasets? How stable are results across different attribution methods and model architectures?
- **Attribution baselines**: Why are only Grad-CAM variants (Grad-CAM++, Layer-CAM, KPCA-CAM, Finer-CAM) considered?
- **Model variety:** Why is the evaluation limited to a single ResNet-18 architecture? Have MSI or LAX been tested on other architectures such as ViTs?
- **LAX generalization and off-manifold behavior**: Have you tested the adapter on out-of-distribution data or with synthetic ground-truth datasets such as FunnyBirds? Was a train/test split used? How does adapter size or architecture affect behavior?
- **Computational cost**: How does the cost of computing MSI and training LAX compare to existing metrics or attribution methods?

---

### Official Review · Reviewer_dzRG · 2025-11-06

**Soundness:** 3
**Presentation:** 3
**Contribution:** 2
**Rating:** 4
**Confidence:** 3

**Summary:**

The paper tackles how to quantify the quality of visual explanations when there’s no ground-truth saliency to compare against. It introduces MSI (Minimality–Sufficiency Integration)—a perturbation-based metric inspired by the information bottleneck—that rewards explanations that are both specific (sufficient) and compact (minimal), by contrasting show/hide perturbation curves and integrating a mask-size penalty. In tandem, it proposes LAX (Learnable Adapter eXplanation), a lightweight adapter trained on top of a frozen model to generate saliency maps without ground-truth explanation labels. Experiments on Synthetic MNIST, CIFAR-10, and CUB-200 illustrate the metric and method working together, with tables and examples reported across these datasets. Finally, the paper outlines extensions to multimodal data and large transformers.

**Strengths:**

Proposes MSI, a clean metric that combines sufficiency and compactness into a single criterion. By integrating the show/hide perturbation curves with a mask-size penalty, it’s better suited to handling cases with multiple plausible supporting regions.

The metric is more interesting than the method: MSI has real discussion value and, if adopted, could help reduce the inconsistencies often seen across insertion/deletion-style evaluation metrics.

On the training side, LAX is relatively lightweight: it does not require saliency annotations, can be attached to a frozen backbone, and is straightforward to deploy in practice.

Experiments on synthetic MNIST, CIFAR, and CUB subsets are largely consistent with the paper’s claims: in multi-solution scenarios, MSI rankings align better with intuition.

**Weaknesses:**

Hyperparameter sensitivity: MSI depends quite heavily on the choice of α_min and the mask penalty (with different values used for different datasets). Method rankings may change with these settings, and the paper does not provide a robustness analysis over these choices.

Perturbation strategy under-specified: the choice of insertion/deletion “fill-in” or reconstruction strategy is not clearly fixed or ablated, even though it can significantly affect the curves and introduce distribution shift, which in turn affects MSI.

Limited baselines: important and commonly used methods such as RISE, Score-CAM, Integrated Gradients, LRP, and SmoothGrad are missing, which weakens the generality of the empirical conclusions.

“Black-box” claim is overstated: LAX relies on intermediate feature maps and additional forward passes, which is closer to a frozen white-box than a true API-only black-box setting.

**Questions:**

Please see Weakness part.

---

### Meta-Review · Area_Chair_dMPH · 2025-12-25

**Summary:**

The reviewers found the simplicity of implementation (Reviewer gg79) of proposed method, the focus on sufficiency and minimality on a metric (Reviewers dzRG & gg79), and the no need of ground-truth annotations strong properties of the proposed method.

On the other hand, there were shared concerns regarding the empirical validation of the proposed methods as standard baselines were not considered and in some cases the models and data modalities that were included were too limited. Very related, there were concerns (Reviewers dzRG & gg79) on the sensitivity of the proposed methods to hyperparameters as proper analysis seem to be missing. There were also concerns (Reviewer gg79 & Gsqk) regarding the actual novelty of the proposed method as it seems to be a combination of existing perturbation-based methods, in particular with close similarity to XRAI (Kapishnikov et al., ICCV 2019). In this regard, proper positioning w.r.t. existing efforts would had strengthened the manuscript.

Some of these are major concerns, and considering that no rebuttal was provided it is hard to recommend the acceptance of this work.

**Reviewer Concerns:**

Addressed Concerns:

- No rebuttal was provided, none of the raised concerns were addressed

Outstanding

- No rebuttal was provided for the concerns raised by the reviewers. All the concerns remain.

**Reviewer Scores:**

As no rebuttal was provided and the reviews pointed to major issues, I do not believe a positive change in score would had been possible without a significant amount of work which might had led to the rewriting of the paper.

---

### Decision · Program_Chairs · 2026-01-26

Reject